Neuroimaging 'will to fight' for sacred values: an
empirical case study with supporters of an
Al Qaeda associate. *R. Soc. open sci.* **6**: 181585.

neuroscience/behaviour/psychology

sacred values, will to fight, neuroimaging, fMRI,
radicalization, violent extremism

**Authors for correspondence:**
Scott Atran
e-mail: satran@umich.edu
Oscar Vilarroya
e-mail: oscar.vilarroya@uab.cat

[†]These authors contributed equally to this work.

Electronic supplementary material is available
online at https://dx.doi.org/10.6084/m9.figshare.
c.4502537.

# Neuroimaging 'will to fight' for sacred values: an empirical case study with supporters of an Al Qaeda associate

Nafees Hamid[1,2,†], Clara Pretus[1,3,†], Scott Atran[1,4,5,6],
Molly J. Crockett[7], Jeremy Ginges[1,8], Hammad Sheikh[1,8],
Adolf Tobeña[1,3], Susanna Carmona[9,10], Angel Gómez[1,11],
Richard Davis[1,4,12] and Oscar Vilarroya[1,3,13]

[1]Artis International, 6424 E. Greenway Parkway, Suite 100-498, Scottsdale, AZ 85254, USA
[2]Department of Security and Crime Science, University College London, 35 Tavistock Sq.,
Kings Cross, London WC1H 9EZ, UK
[3]Departament de Psiquiatria i Medicina Legal, Universitat Autònoma de Barcelona, 08193
Cerdanyola del Vallès, Spain
[4]The Changing Character of War Centre, Pembroke College, University of Oxford, St. Aldates,
Oxford OX1 1DW, UK
[5]Centre National de la Recherche Scientifique, Institut Jean Nicod – Ecole Normale
Supérieure, 29 rue d'Ulm, 75005 Paris, France
[6]Gerald Ford School of Public Policy and Institute for Social Research, University of Michigan,
Ann Arbor, MI 48109, USA
[7]Department of Psychology, Yale University, 2 Hillhouse Ave, New Haven, CT 06511, USA
[8]Department of Psychology, New School for Social Research, 80 5th Ave, New York, NY
10011, USA
[9]Centro de Investigación Biomédica en Red de Salud Mental, Madrid, Spain
[10]Unidad de Medicina y Cirugía Experimental, Instituto de Investigación Sanitaria Gregorio
Marañón, Madrid, Spain
[11]Departamento de Psicología Social y de las Organizaciones, Universidad Nacional de
Educación a Distancia, UNED, C/Juan del Rosal, No. 10, 28040 Madrid, Spain
[12]School of Politics and Global Studies, Arizona State University, Coor Hall, 975 S. Myrtle
Ave., Tempe, AZ 85287, USA
[13]IMIM, Hospital del Mar Medical Research Institute, Passeig Marítim 25-29, Barcelona
08003, Spain

SA, 0000-0002-0796-7279; OV, 0000-0001-8285-5624

Violent intergroup conflicts are often motivated by
commitments to abstract ideals such as god or nation, so-
called 'sacred' values that are insensitive to material trade-
offs. There is scant knowledge of how the brain processes
costly sacrifices for such cherished causes. We studied
willingness to fight and die for sacred values using fMRI in

Barcelona, Spain, among supporters of a radical Islamist group. We measured brain activity in radicalized individuals as they indicated their willingness to fight and die for sacred and non-sacred values, and as they reacted to peers' ratings for the same values. We observed diminished activity in dorsolateral prefrontal cortex (dlPFC), inferior frontal gyrus, and parietal cortex while conveying willingness to fight and die for sacred relative to non-sacred values—regions that have previously been implicated in calculating costs and consequences. An overlapping region of the dlPFC was active when viewing conflicting ratings of sacred values from peers, to the extent participants were sensitive to peer influence, suggesting that it is possible to induce flexibility in the way people defend sacred values. Our results cohere with a view that 'devoted actors' motivated by an extreme commitment towards sacred values rely on distinctive neurocognitve processes that can be identified.

# 1. Introduction

Rational choice theories generally assume that individuals choose among available options, weighing up the potential costs and benefits of each option as well as their likelihoods [1]. But such theories may be inadequate to explain or predict willingness to fight and die for a cause if such willingness is shaped by a devotion to sacred values and group identities that people are fused with [2,3]. Sacred values are preferences, beliefs and practices that communities deem protected from material trade-offs [4], as when land or law becomes holy or hallowed [5], however materially advantageous such values may prove in the evolutionary long run [6].

The construct of sacred values has accumulated a compelling corpus of findings [4,7] and has been studied in the context of deep-seated political conflicts [5,8–11]. Field studies and experiments with populations involved in armed conflict find that self-reporting of support for violence appears insensitive to material costs and benefits [12–14], and asking people to trade sacred values for material benefits provokes moral outrage [5,13]. Highly committed activists, 'devoted actors', are willing to sacrifice self-interests, even at the cost of their lives and family, for sacred in-group values [5,12–14]. This feature of intergroup conflict, where people fight on when odds of victory are low, suggests choices made independently of calculated risks and likely outcomes. If so, then a primary focus on understanding, preventing or deterring such behaviours through utilitarian cost-imposition strategies [15] may be insufficient.

One question that is not easily addressed with behavioural studies alone is the extent to which costly decisions involving sacred values are mediated by cost–benefit evaluations [16–19]. In this sense, our first motivation to use neuroimaging on radicalized individuals was to detect underlying processes that can provide potentially useful insights about their costly preferences, in particular the role that such preferences may play in determining will to fight. Second, we sought to discover whether sacred values and non-sacred values are processed by similar or distinct neural circuits, and to describe which circuits or processes might be involved. Finally, we attempted to validate behavioural modulation effects on value commitment by factors that differentially affect sacred and non-sacred values, and to identify neuro-cognitive signatures for these effects.

Previous work has identified brain regions implicated in utilitarian cost–benefit calculations during value-based decision-making. Specifically, trading off costs and benefits—both in simple decisions involving outcomes for oneself as well as more complex moral decisions—has been associated with neural activity in subcortical regions, such as striatum and amygdala, as well as cortical regions including lateral prefrontal cortex, ventromedial prefrontal cortex (vmPFC) and posterior parietal cortex [20,21]. The vmPFC, in particular, has been hypothesized to function as a global value comparator during value-based choices, and its activity has been found to correlate with value differences between chosen and unchosen items in a variety of contexts, from monetary settings up to food preferences [20,22]. In addition, prior studies suggest that the dorsolateral prefrontal cortex (dlPFC) modulates vmPFC activity during decisions that require cognitive control, such as sticking to healthy food choices [23], or reweighting of stimulus in front of new contextual information in monetary settings [24].

This prior work allowed us to specify our hypotheses. If processing sacred values involves a reduced reliance on cost–benefit utilitarian calculations, as anticipated by a devoted actor framework and behavioural research on sacred values [2,8,11,12,14], then existential decisions about sacred values relative to non-sacred values, as in willingness to fight and die, should involve less activity in areas previously implicated in cost–benefit computations. To be able to test this, we studied willingness to

1. Willingness to fight and die for sacred values: intra-scanner paradigm, 80 trials, 2 runs, 16.7 min

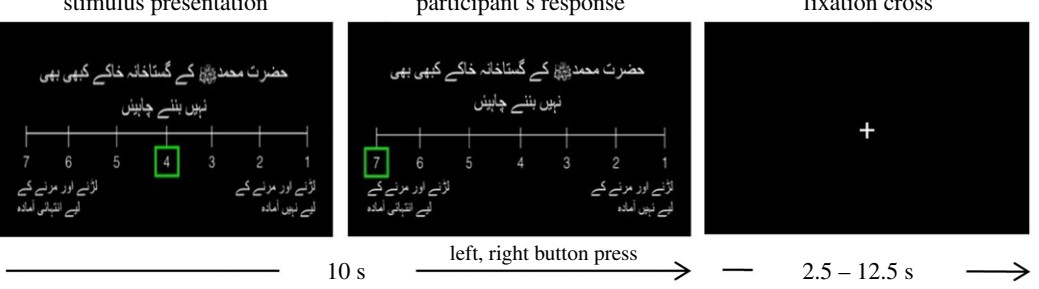

| stimulus presentation | participant's response | fixation cross |
| --- | --- | --- |
| | left, right button press | |
| 10 s | | 2.5 – 12.5 s |

2. False consensus feedback for willingness to fight and die: intra-scanner paradigm, 120 trials, 2 runs. 18.75 min

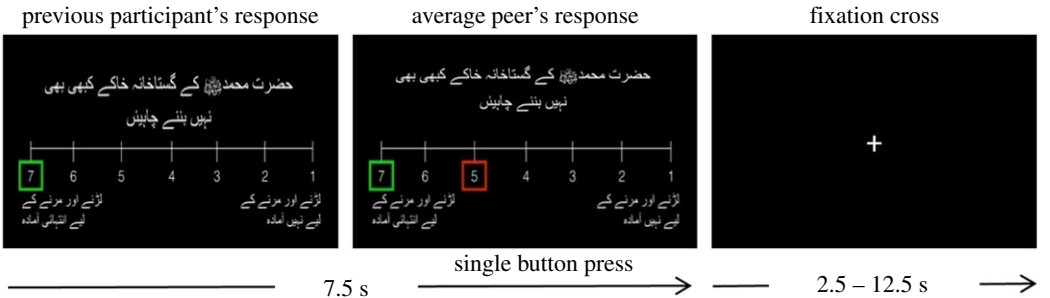

| previous participant's response | average peer's response | fixation cross |
| --- | --- | --- |
| | single button press | |
| 7.5 s | | 2.5 – 12.5 s |

3. Post-scanning session: post-manipulation willingness to fight and die rating + emotions questionnaire

**Figure 1.** Scheme of the overall testing timeline including trial structure in Rating 1 and Feedback sessions. Trials are exemplified with a screenshot of one of the items presented in the Urdu version ('Prophet Mohammed must never be caricatured') with a Likert scale ('Not at all willing to fight and die' to 'Extremely willing'). Rating 1 trials began with the cursor centred in 4 (marked in green), which participants moved along the Likert scale to convey their rating. Feedback trials began with previous participants' ratings on the same item (framed in green), and community feedback (framed in red) was presented following button press. After Rating and Feedback sessions, a behavioural test session including Rating 2 and the emotions questionnaire was completed outside the fMRI scanner.

fight and die for sacred values using fMRI among supporters of the radical Islamist group *Lashkar-et-Taiba* (Army of the Righteous) recruited from different neighbourhoods in and around Barcelona, Spain. *Lashkar-et-Taiba*, which executed the 2001 attack on India's Parliament and the 2008 Mumbai attacks, has been designated a terrorist organization by the USA, European Union and Russia. UN Security Council Resolution 1267 (May 2008) considers *Lashkar-et-Taiba* 'an entity associated with Al Qaeda'.

After a long selection process based on extensive ethnographic work and field surveys, we invited those candidates expressing more radicalized political views, including support for *Lashkar-et-Taiba*, to the fMRI facilities. There, we measured the candidates' brain activity while they indicated their willingness to fight and die for sacred and non-sacred values (figure 1 (1)). We chose Spain, a recurrent target of jihadists vowing to recover Al-Andalus, Western Europe's last Muslim polity, which fell in 1492. This was an explicit motivation for the 2004 Madrid train bombings, Europe's worst terrorist attack, along with grievances pertaining to Western involvement in ongoing conflicts in Muslim lands, as with the August 2017 attack in Barcelona that occurred subsequent to our study.

We also explored a second issue related to potential interventions with respect to willingness to fight and die for sacred values. Although behavioural work suggests willingness to fight and die for sacred values is relatively insensitive to cost–benefit reasoning, it may be possible to modulate it using methods that do not entail material incentives or threats. In our second fMRI experiment we investigated one such strategy: influencing via peer feedback. Ever since Solomon Asch's [25] pioneering work on social conformity, extensive research in social cognition has shown that human behaviour is not only guided by subjective attitudes and values but also by perceptions of others' attitudes and values [26]. Some studies have shown that inducing false perceptions of others' moral judgements can influence people to change their own moral judgements [27,28]. Research on sacred values, however, has shown that such values are resistant to social influence [8]. Neuroimaging research suggests that when social consensus conflicts with subjects' judgements, the degree of activation in prediction error networks and orbitofrontal cortex predicts the degree of change in judgements at a later

time [29,30]. The only neuroimaging study published on social conformity and sacred values to-date revealed that increased ventrolateral prefrontal cortex (vlPFC) activity for sacred values predicted weaker social conformity, whereas decreased vlPFC activity predicted greater conformist behaviour [31].

Research on radicalization distinguishes between deradicalization and disengagement, suggesting that former violent extremists rarely change their beliefs (deradicalize) but more often lose their motivation to defend them (disengage) [32]. Accordingly, we conjectured that it might be possible to induce flexibility in the way people defend their sacred values. Thus, we tested whether willingness to fight and die for sacred values is amenable to social influence and, if so, whether regions previously associated with attitude change [29,30] correlate with change in judgement for sacred versus non-sacred values. For this, we used a false consensus feedback paradigm on the same radicalized sample. Participants were first reminded of their previous responses on willingness to fight and die for sacred and non-sacred values. Then, they were presented higher, equal or lower willingness to fight and die ratings representing the average opinion of their in-group, described as 'the Pakistani community in Barcelona' (figure 1 (2)).

A final caveat is in order. Insofar as our aim was to uncover different decision pathways mediating willingness to fight and die for sacred values compared with non-sacred values in the same individuals, we purposely sought a sample where extreme attitudes and associated behaviours were organized and connected to action: namely, participants with radicalized religious and political leanings. Accordingly, we did not include any one of many potential control groups because our interest was to comprehend what might drive members of a radicalized group to act violently for a cause, not what may generally distinguish radicalized from non-radicalized individuals (an important, but different, research topic). When confronted with violent actions from members of a radicalized group the immediate practical problem—whether for police, military, or the public at large—is to know about and deal only with members of a radicalized group. On a theoretical plane, knowing that behaviour related to sacred values in general, and to violent actions in particular, systematically differ from behaviours and actions related to non-sacred values in ways that rule out posturing has potentially significant implications for understanding non-utilitarian dimensions of human decision making and the drivers of violent conflict.

# 2. Material and methods

## 2.1. Experimental design

In order to compare neural activity between sacred and non-sacred values using fMRI, we developed a list of candidate values from fieldwork, involving topics ranging from Indian rights over Kashmir to caricatures of the Prophet Muhammad. Prior testing determined which values were sacred and non-sacred for each participant. In line with previous work on sacred values [5,7,8,33], a value was considered sacred if participants refused any material incentive to give it up, and if their stance showed immunity to social influence. Participants ranked the importance of each value. For every participant we selected the top-ranked six sacred values, along with six non-sacred values chosen randomly. While undergoing brain scanning using fMRI, participants rated their willingness to fight and die for each value (figure 1 (1)). Linguistic variations of each value produced 80 different trials in order to increase the number of trials within the fMRI paradigm without wording repetitions.

In order to investigate potential modulatory effects of social influence on willingness to fight and die for sacred values we used a false consensus feedback paradigm [29] (figure 1 (2)). In it, we presented participants with conflicting (peers-lower), non-conflicting (peers-agree) and peers-higher willingness to fight and die average ratings of the participants' peers for sacred and non-sacred values, presented as 'the average opinion of Barcelona's Pakistani community'. After the fMRI session participants were asked again to rate their willingness to fight and die for each value, in order to identify possible changes in judgement owing to peer feedback. Then participants were asked to convey their emotions (e.g. moral outrage) at witnessing peer feedback for every item (electronic supplementary material, figure S5).

## 2.2. Participants

To select participants and design culturally relevant stimulus materials, we conducted ethnographic fieldwork from early 2014 through 2015 with the Pakistani immigrant community in Barcelona to understand this community's general cultural context (including political and religious sensitivities).

Exclusion criteria for the fMRI study included presence of psychiatric diseases or personality disorders, assessed by the MINI International Neuropsychiatric Interview.

Through participant observation and in-depth interviews we sought to identify appropriate respondents for an fMRI study of supporters of militant jihad and their willingness to make costly sacrifices for their values (see electronic supplementary material, section S1.1). Based on this fieldwork we recruited a subset of 30 male respondents who scored high on a construct representing political radicalization (see electronic supplementary material, section S1.3). All supported the militant group *Lashkar-et-Taiba*, which UN Security Council Resolution 1267 (May 2008) designated 'an entity associated with Al Qaeda'.

The total ethnographic sample consisted of male respondents, $n = 146$, average age $M = 30.82$ years (range 18 to 62). All respondents were Muslim, and most self-identified as Sunni. As the goal of fieldwork was to identify supporters of militant global jihadism for the fMRI study, we applied a series of increasingly strict criteria related to degree of individual radicalization. In addition to (i) agreement with jihadism and (ii) approval of violence against civilians, we only considered respondents for the fMRI study who also expressed (iii) willingness to personally advocate or use violence for jihad (see electronic supplementary material, section S1.2, for complete list): to engage in violent protest ($n = 110$), to join a non-state militant group ($n = 27$) or to fight and die on their own ($n = 51$). From those $n = 45$ respondents who met all three criteria, $n = 30$ agreed to participate in the fMRI study. (For details of the stages in the field study, pre-selection field survey and selection measures, and comparison of fMRI participants with other respondents, electronic supplementary material, sections S.1–S.5).

## 2.3. Neuroimaging data acquisition

Images were acquired in a Siemens 3T scanner. T1-weighted images were obtained using a FSPGR sequence (TR: 11.6 ms, TE: 4.8 ms, FA: 12, matrix size: $280 \times 280$, 150 slices, slice thickness: 1.00 mm). An EPI-T2* sequence allowed obtaining the functional volumes, each comprising forty 3.4 mm thick slices (TR: 2500 ms, TE: 27 ms, FA: 90, matrix size: $64 \times 64$, 40 slices, slice thickness: 3.4 mm).

SPM12 (Wellcome Department of Imaging Neuroscience, London, United Kingdom) and the Analysis of Functional Neuroimaging software [34] were employed for the functional MRI data analysis. Functional images were despiked (AFNI), corrected for motion-related artefacts (SPM), normalized to MNI standard space (SPM), smoothed with a 8 mm full-width-at-half-maximum Gaussian kernel (SPM) and detrended (AFNI).

## 2.4. Neuroimaging paradigm

The complete neuroimaging session included two fMRI paradigms (Rating 1 and Feedback) followed by a behavioural testing session where participants completed Rating 2 and an emotions questionnaire (figure 1).

The rapid event-related fMRI paradigm was designed using Matlab 2012a (The MathWorks, Inc., Natick, Massachusetts, United States) with Psychtoolbox extensions. Before starting the fMRI paradigms, participants practised the task on a computer until fully comprehending the exercise and mastering control of the response buttons. Different grammatical formulations for each item were included in order to avoid habituation effects (see below).

Rating 1 encompassed 80 trials split in two runs (electronic supplementary material, table S1). From the list of sacred and non-sacred values elaborated for each participant, the six top sacred values ranked in order of importance by the participant were included, while six non-sacred values were selected randomly. A list of seven grammatical rephrasings for each candidate value was carefully developed by an Urdu linguist and revised by a group of Urdu speaking collaborators. Rephrasings for the selected 12 values were included to produce different items for each of the 80 trials. Thus, each participant completed an individualized version, including their own collection of sacred and non-sacred values.

The Feedback section included 120 items split in two runs, encompassing the 80 items presented in Rating 1 and 40 randomly chosen repetitions of the same values. In this block, participants witnessed their own previous rating framed in green followed by what was presented as 'the average opinion of the Pakistani community in Barcelona' framed in red. To keep participants attentive, they were instructed to press a button to proceed to the community feedback screen. Community ratings were manipulated to be 2 points less willing to fight and die (peers-lower condition), non-conflicting with the participant (peers-agree condition) and 2 points more willing to fight and die (peers-higher condition) in the same proportion of trials in order to obtain enough statistical power, that is,

approximately 30 trials for each of the 3 feedback conditions × 2 sacredness conditions (sacred values/non-sacred values). Each value received consistent feedback in all rephrased forms. Extreme ratings could only be included either in the peers-agree condition or in one of the conflicting conditions (peers-higher for Ratings 1 and 2 and peers-lower for Ratings 6 and 7). Given that most of the time sacred value scores were close to ceiling (mean = 6.59 out of 7 points, s.d. = 0.49), the peers-higher condition could only be rarely implemented for sacred values. Of 30 participants, only four received peers-higher feedback in at least one sacred value. For this reason, the peers-higher condition was dropped from the behavioural and neuroimaging analyses.

Immediately after the scanning session, and without previous notice, participants completed a third task, Rating 2, where they had to convey willingness to fight and die for each of the 12 values included in the scanning paradigm to detect possible changes of judgement (rephrasings for values were excluded at this point).

In the final stage participants completed an emotions questionnaire, where they were first reminded of their own ratings in Rating 1 along with their peers' ratings in the Feedback section. Then, they were asked to what degree (from 1 to 7 points) they experienced a series of emotions at witnessing the Pakistani community's normative opinion, including disgust, anger, contempt, shame, joy, compassion and pride (see electronic supplementary material, figure S5). Disgust, anger and contempt scores were collapsed into a 'moral outrage' construct.

## 2.5. Statistical analysis

The GLM matrix for the willingness to fight and die paradigm included 2 sacredness regressors (sacred/non-sacred values) each including a parametric modulator for willingness to fight and die ratings and a parametric modulator for reaction time. In addition, a button press regressor and 6 movement regressors were included as regressors of no interest. Contrasts of interest included neural activity during (i) the sacred value compared with the non-sacred value condition and (ii) neural activity predicted by willingness to fight and die ratings for sacred values as a parametric modulator.

The GLM matrix for the Feedback paradigm encompassed 6 regressors (3 feedback types × 2 sacredness conditions) and 6 movement regressors. Inasmuch as sacred value generally obtained close to ceiling scores, the 'peers-higher' feedback condition could not be submitted to second-level analyses. To capture neural activity associated with change in judgement, a second-level regression analysis was conducted with change in judgement used as predictor for neural activity in the peers-lower feedback condition for sacred versus non-sacred values.

In each contrast, a whole-brain analysis was conducted using a threshold of $p < 0.05$ corrected for multiple comparisons by means of the family-wise error rate (FWE) at a cluster level, with a peak-level threshold of $p < 0.001$. One subject had to be removed from the analysis owing to technical problems.

## 2.6. Controlling for confounding effects

Because sacred and non-sacred values potentially differ along a number of dimensions unrelated to sacredness *per se*, we conducted a number of control analyses to rule out the possibility that the observed neural differences between processing sacred and non-sacred values could be explained by factors such as emotional intensity, familiarity and salience. We also used this survey to check sacred value persistence by retesting all values for sacredness. Six months after the fMRI study, participants rated the sacredness as well as the emotional intensity, familiarity and salience of each of the fMRI stimuli. As some of these participants had left the country by the time of the follow-up data collection, we were successfully able to reach 18, all of whom agreed to participate in the follow-up survey.

In this survey, we obtained the following measures for each value that had been presented to participants in the fMRI study: value sacredness (monetary trade-off resistance and reluctance to accept democratic consensus over the opposite position), emotions ('How much of each emotion do you feel when you think about defending this issue?' including anger, joy, excitement, fear, sadness), familiarity ('I am quite familiar with this issue' and 'Comparing with others, I think I am quite familiarized with this issue'), frequency ('How frequently do ideas, thoughts, impulses, images related to [sacred value] occur?'), and mental energy ('How much of your mental energy/attention do these ideas, thoughts, impulses, or images related to [sacred value] take up?'). These new measures were included in complementary behavioural and neuroimaging analyses, as described below.

As for the behavioural analyses, a chi-square test for association was conducted between the main study sacred value rating and the follow-up sacred value rating. Ratings in emotions (anger, joy, excitement, sadness and fear), familiarity, frequency and mental energy for sacred and non-sacred values were calculated for each participant and submitted to a series of paired-sample $t$-tests comparing sacred and non-sacred value scores in each emotion, familiarity, frequency and mental energy. Anger, excitement and joy scores were averaged into an overall 'emotional intensity' score (Cronbach's alpha = 0.961). The two familiarity items were averaged to create a new familiarity construct (Cronbach's alpha = 0.80) and frequency and mental energy scores were averaged into an overall 'salience' score, as these were significantly correlated ($r = 0.775$, $p < 0.001$). As shown in electronic supplementary material, figure S6, moderate to strong correlations were found between willingness to fight and die from the main study and the new measures collected in the follow-up (emotional intensity: $r = 0.535$, $p < 0.001$; familiarity: $r = 0.455$, $p < 0.001$; frequency: $r = 0.508$, $p < 0.001$; mental energy: $r = 0.568$, $p < 0.001$).

Complementary neuroimaging analyses involved three different GLMs including trial-by-trial emotional intensity, familiarity and salience scores, respectively, as first-level parametric regressors for the sacred and the non-sacred value conditions, together with the reaction time regressors. These regressors were built by retrieving which particular values had been presented to each participant in each trial (the paradigm included individualized batteries of values for each participant and trials had been randomized) and concatenating each of the value scores in emotional intensity, familiarity and salience for sacred and non-sacred values according to value presentation order, reflecting trial-by-trial scores in each of the confounds. The regressors were then included in three different first-level GLMs together with the sacred value, non-sacred value and button press regressors, the parametric regressors for reaction time and 6 movement regressors. Each of the three first-level GLMs were submitted to a one-sample $t$-test to evaluate neural activity in the sacred versus non-sacred value, once activity associated with emotional intensity, familiarity and salience, respectively, had been accounted for by the first-level parametric regressors (see electronic supplementary material, figure S7A).

As a second, more conservative control, we sought to mask out from our original sacred versus non-sacred value contrast regions associated with emotional intensity, familiarity and salience at a liberal threshold ($p < 0.01$, uncorrected; electronic supplementary material, figure S7). Activity correlated with these three confounds was binarized and used as mask to exclude activity correlated with emotionality, familiarity and salience from the non-sacred value greater than sacred value contrast in the original dataset ($N = 29$). Brain regions included in these masks encompassed the insula, the anterior and middle cingulate cortex, the caudate and the precuneus (see masks in electronic supplementary material, figure S8B).

Finally, we turned to the broader literature to search for regions implicated in emotional intensity, familiarity and salience that could account for the observed differences in activity between sacred and non-sacred value conditions. For this purpose, we used Neurosynth [35], a platform that calculates probability maps reflecting the likelihood that a term such as 'emotion' has been used in a study based on the activation in a given voxel by evaluating 11 406 different studies. Thereby, we extracted three different masks encompassing brain areas positively and negatively correlated with 'emotion', 'familiarity' and 'salience', respectively, thresholded at $Z = 5$ (see electronic supplementary material, figure S8C). The resulting six masks (three from the follow-up analysis and three from Neurosynth) were used to exclude activity associated with emotions, familiarity and salience from the non-sacred value greater than sacred value contrast obtained from the original GLM, with no significant alteration of results (see electronic supplementary material, figure S7B and C).

# 3. Results

## 3.1. Experiment 1

Willingness to fight and die ratings were substantially higher for sacred values (mean = 6.61 out of 7 points, s.d. = 0.48) than for non-sacred values (mean = 3.8, s.d. = 1.31; comparison: $t_{29} = 11.93$, $p < 0.001$, $\eta^2_{partial} = 0.831$). Willingness to fight and die ratings were also conveyed faster in trials comprising sacred values (mean = 4.72 s, s.d. = 1.49) compared with non-sacred values (mean = 5.49, s.d. = 1.29; comparison: $t_{29} = -3.52$, $p = 0.001$, $\eta^2_{partial} = 0.299$).

Emotional intensity, familiarity and salience ratings were also higher for sacred compared with non-sacred values (mean differences in ratings between sacred and non-sacred values: anger = 4.47, joy = 3.24,

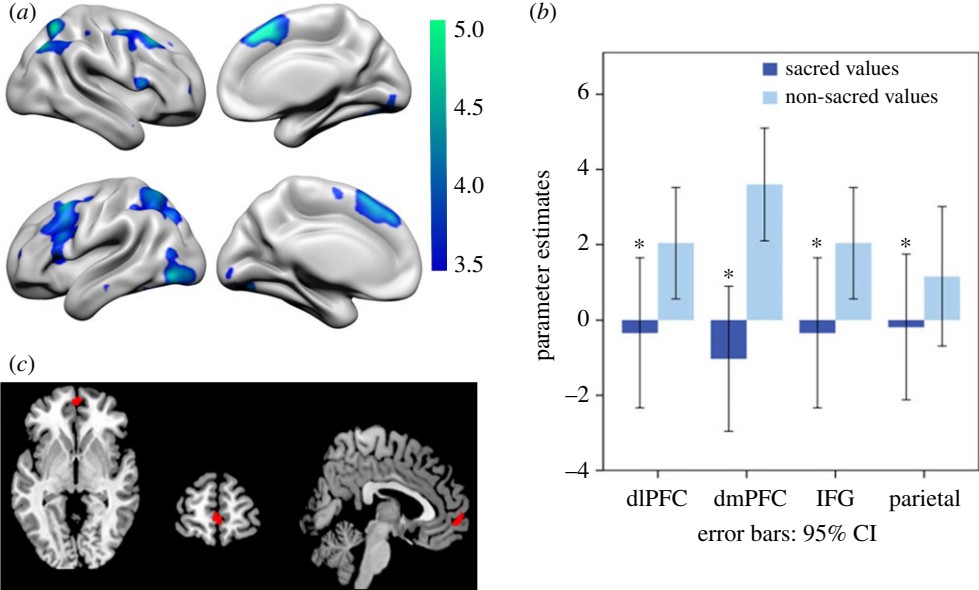

**Figure 2.** Neural signatures with willingness to fight and die for sacred values. (*a*) Activity in the dorsolateral prefrontal cortex (dlPFC), the inferior frontal gyrus (IFG) and the superior parietal cortex was decreased during sacred compared with non-sacred value assessment ($T = 3.40$, $p < 0.05$ FWEc, single voxel $p < 0.001$). The colour bar represents $t$-values. (*b*) Parameter estimates of the clusters of activation in the dlPFC, dorsomedial prefrontal cortex (dmPFC), IFG and parietal lobule (binary masks extracted from the non-sacred values greater than sacred values contrast) were negative and significantly lower for the sacred values greater than baseline contrast compared with the non-sacred values greater than baseline contrast (marked with *, dlPFC: Wilk's lambda = 0.7951, $F_{1,28} = 7.21$, $p < 0.012$; dmPFC: Wilk's lambda = 0.661, $F_{1,28} = 14.33$, $p < 0.001$; IFG: Wilk's lambda = 0.7951, $F_{1,28} = 7.21$, $p < 0.012$). (*c*) During ratings, vmPFC showed a positive parametric response to willingness to fight and die for sacred values (small volume correction, $T = 3.40$, $p < 0.05$ FWEc, single voxel $p < 0.001$, in red).

excitement = 3.23, familiarity = 1.05, frequency = 1.44, mental energy = 1.35, all $p < 0.001$, see electronic supplementary material, figure S6). Thus, analyses of neuroimaging results below controlled for these measures. Value sacredness was stable after six months ($\chi^2_1 = 175.46$, $p < 0.001$; $\varphi = 0.901$, $p < 0.001$).

At a neural level, the sacred value condition elicited less activity in the bilateral dlPFC, inferior frontal gyrus (IFG), superior and inferior parietal cortex, and right caudate than the non-sacred value condition (figure 2*a* and electronic supplementary material, table S2; all analyses were thresholded at $T = 3.40$, $p < 0.05$ FWEc, single voxel $p < 0.001$, parameter estimates shown in figure 2*b*). No voxels were significantly more active in the sacred value condition relative to the non-sacred value condition.

Because RTs were different between conditions and have been associated with brain regions that partly overlap with those in our findings [35], we controlled for trial-by-trial reaction times (RTs) and masked out neural activity predicted by the parametric RT regressor with RTs (see mask in electronic supplementary material, figure S8A), leaving results unaffected. Results also survived after controlling for emotional intensity, familiarity, or salience trial-by-trial ratings (dlPFC, dmPFC and parietal cortex, $T = 3.64$, $p < 0.05$ FWEc, single voxel $p < 0.001$, see electronic supplementary material, figure S7A), masking out neural activity associated with these three confounds (dlPFC and parietal cortex, $T = 3.4$, $p < 0.05$ FWEc, single voxel $p < 0.001$, see electronic supplementary material, figure S7B), and masking with Neurosynth [35] masks for 'emotion', 'familiarity' and 'salience' (see electronic supplementary material, figure S7C). Meanwhile, activity within a mask created based on a Neurosynth search of the term 'calculation' showed significant differences between the sacred and non-sacred value conditions in the left dlPFC, left dmPFC, left IFG and bilateral parietal cortex ($T = 3.4$, $p < 0.05$ FWEc, single voxel $p < 0.001$).

Willingness to fight and die across all conditions (sacred and non-sacred values) did not significantly predict positive activity in any voxel, although relaxing the threshold to $p < 0.005$ revealed a cluster in vmPFC (small volume correction with willingness to fight and die for sacred value mask, $T = 2.80$, single voxel $p < 0.005$ uncorrected).

When considering sacred and non-sacred value conditions separately, willingness to fight and die for sacred values correlated positively with vmPFC activity ($p < 0.05$, small volume corrected;

figure 2*c*/upper image). Willingness to fight and die for non-sacred values did not correlate significantly with vmPFC activity. Directly contrasting willingness to fight and die for sacred relative to non-sacred values did not reveal any significant clusters.

In sum, *Lashkar-e-Taiba* supporters expressed greater willingness to fight and die and responded more quickly to sacred than non-sacred values. More importantly, the sacred value condition, compared with the non-sacred value condition, involved less activation in neural areas previously associated with cognitive control and utilitarian reasoning, namely, dlPFC, IFG and parietal cortex [36,37]. This difference in neural activation between sacred and non-sacred values was robust when controlling for reaction times, emotional intensity, familiarity and salience. This is not to deny that emotions and other factors may be integral to sacred values [5]; however, our approach sought to ensure that sacredness could not be attributed exclusively to these other factors.

Neural activity differences between sacred and non-sacred values also cannot be readily attributed to differences in how well-formed participants' opinions on sacred values were compared with non-sacred values; for, inasmuch as the surveys were specifically designed to ensure that all values (including sacred and non-sacred values) were important and common issues for the community, in principle all participants had a well-formed opinion about them.

## 3.2. Experiment 2

For both sacred and non-sacred values there was a significant change in willingness to fight and die ratings in the direction established by peers after participants received conflicting (peers-lower) community feedback (Wilk's lambda $= 0.783$, $F_{1,29} = 8.03$, $p = 0.008$, $\eta^2_{partial} = 0.217$) with no statistical interaction with value sacredness (sacredness $\times$ feedback $\times$ time interaction: Wilk's lambda $= 0.994$, $F_{1,29} = 0.183$, $p = 0.672$, see figure 3*b*). In addition, the sacred value condition evoked higher degrees of both moral outrage (built as an average of anger, contempt and disgust scores, Cronbach's alpha $= 0.955$) and joy at peers' willingness to fight and die ratings compared with the non-sacred value condition (figure 3*c*). Specifically, reported post-manipulation moral outrage ratings were substantially higher after conflicting/peers-lower feedback (Wilk's lambda $= 0.559$, $F_{1,29} = 22.92$, $p < 0.001$, $\eta^2_{partial} = 0.463$) and joy ratings were higher after non-conflicting/peers-agree feedback (Wilk's lambda $= 0.765$, $F_{1,29} = 8.27$, $p = 0.008$, $\eta^2_{partial} = 0.235$) when values were sacred compared with non-sacred.

We found that change in willingness to fight and die for sacred values as a second-level regressor was correlated with activity in the right dlPFC upon receiving conflicting feedback for sacred compared with non-sacred values (small volume correction with non-sacred value greater than sacred value mask, $T = 3.42$, $p < 0.05$ FWEc, single voxel $p < 0.001$, see figure 3*a*). This region overlapped with the region of dlPFC identified in the contrast between expressing willingness to fight and die for sacred versus non-sacred values ($T = 3.42$, $p < 0.05$ FWEc, single voxel $p < 0.001$).

Although adherence to sacred values has been shown to be resistant to social influence [8,31], our data suggest that specific behavioural commitments (willingness to fight and die) to support sacred values may be flexible. Indeed, we found that change in willingness to fight and die for sacred values was correlated with activity in the right dlPFC upon receiving conflicting feedback for sacred compared with non-sacred values. This finding suggests that attitude change after conflicting peers' feedback could be mediated by evaluative processes including reweighting of the stimulus [24]. Moreover, conflicting in-group opinions were associated with higher self-reported moral outrage scores which, in turn, predicted insula responsiveness. This observation is consistent with the role of the insula in social aversion [38], including reactions of disgust and indignation [39]. Nevertheless, the moderating effect of social influence on willingness to fight and die was independent of moral outrage, suggesting that social influence may affect commitment to willingness to fight and die in an implicit way.

The fact that willingness to fight and die ratings could be modulated using social influence by the in-group may be relevant for de-radicalization strategies using peer-to-peer interventions specifically targeting willingness to fight and die, rather than sacred value adherence.

## 4. Discussion

We investigated willingness to fight and die for sacred values in a sample of *Lashkar-et-Taiba* supporters using functional neuroimaging. Altogether, our data points to the dlPFC, IFG and parietal cortex as key

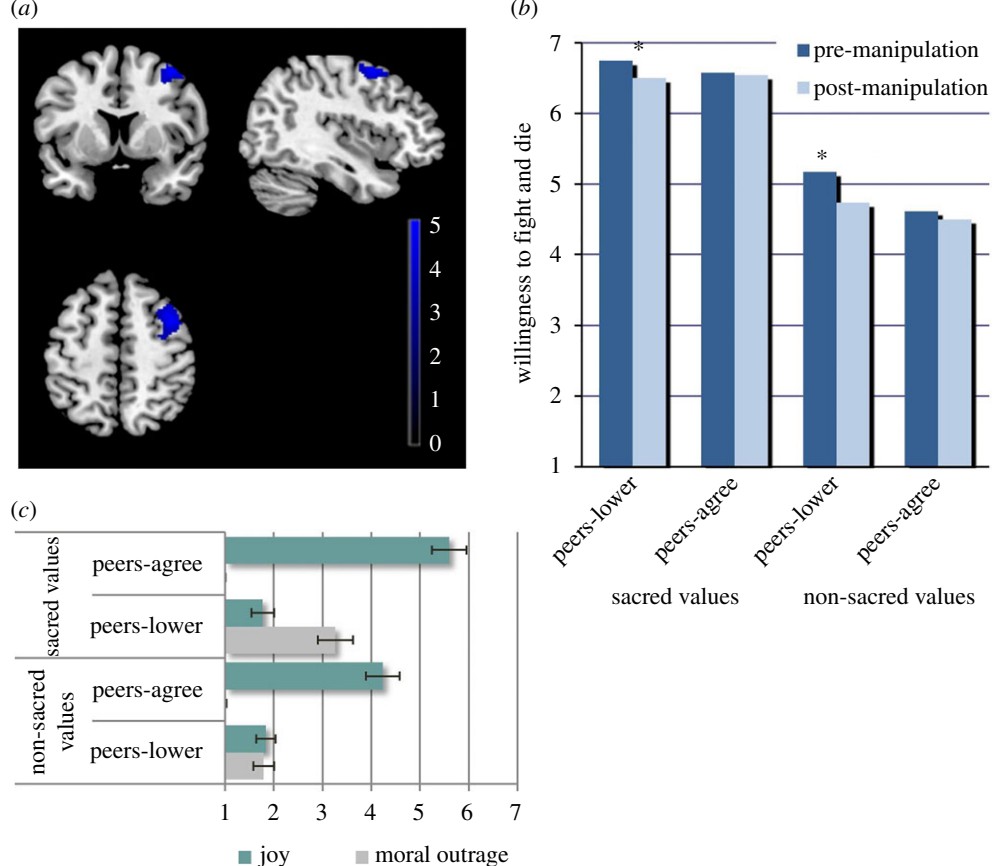

**Figure 3.** Effects of lower willingness to fight and die community ratings (peers-lower) on participants' ratings and self-reported emotions. (*a*) Right dlPFC activity predicted by change in willingness to fight and die for sacred values overlapped with R dlPFC activity associated with non-sacred value versus sacred value during the willingness to fight and die rating paradigm (small volume correction using R dlPFC mask from non-sacred versus sacred value contrast, $T = 3.42$, $p < 0.05$ FWEc, single voxel $p < 0.001$). The colour bar represents $t$-values. (*b*) Significant interaction between feedback type and pre/post-manipulation ratings in willingness to fight and die indicating a change in judgement in alignment with peers ($p < 0.008$). (*c*) Sacred values elicited the greater degrees of emotion compared with non-sacred values, with higher reported moral outrage at conflicting feedback (left, $p < 0.001$) and higher joy at non-conflicting feedback (right, $p < 0.002$). (*) indicates significant results.

regions underlying brain differences between sacred and non-sacred values with regard to decisions about making costly sacrifices, including fighting and dying [14]. At a level of main effects, the bilateral dlPFC was less active during sacred versus non-sacred value decisions, suggesting a decreased reliance on cognitive control functions during sacred value choices. Overall, these observations are consistent with the idea that choices involving sacred values are less dependent on cost–benefit calculations than choices involving non-sacred values, and the view of sacred values as moral imperatives guiding goal-oriented actions [7,40]. They are also consistent with previous literature that addresses the role of utilitarian thinking in moral cognition [37], and with models that distinguish between more cognitively deliberate versus more affectively driven reasoning [41]. In addition, this interpretation dovetails with behavioural data showing that material incentives and disincentives, which can otherwise successfully bias utilitarian reasoning based on cost–benefit calculation, are less effective in influencing behaviour when sacred values are at stake [5,8,10].

The construct of 'Sacred Values' has a relatively long-standing tradition in sociology, anthropology and history [42] and, in the last decades, it has accumulated a compelling experimental corpus in social and political psychology [4,7]. In recent years, this concept has been revisited and enriched by historical and anthropological analyses [11,13] in combination with a series of experiments carried out in the context of deep-seated political conflicts, such as the Israeli–Palestinian conflict [5,8], the Iranian nuclear programme [10], the Muslim–Hindu conflict [9], as well as in other forms of cultural conflict [40]. Sacred values have proven to be behaviourally different from non-sacred values in that people holding them show resistance to instrumental trade-offs, even when the offer is increased to an indefinitely large amount.

Our neuroimaging findings support this distinction, revealing a differential neural activity between sacred values in contrast to (culturally relevant) non-sacred values. We found that sacred value choices involved less activation of brain regions previously associated with cognitive control and cost–benefit calculations [20,21,37]. Figuring out the neural mechanisms that sustain sacred value processing will be key to: (i) validating behavioural modulation effects on value commitment by factors that differentially affect sacred and non-sacred values, and (ii) comparing neural substrates of sacred value processing in different samples with diverse cultural backgrounds in order to define cross-cultural commonalities in sacred value processing.

The question remains of why there were no brain regions associated with affective processing, such as the amygdala, which activated during the sacred compared with the non-sacred value condition. We believe that the most likely explanation for this owes to our experimental paradigm not being sensitive enough to detect the differential neural activity associated with affective regions.

The finding that non-sacred compared with sacred value processing relied on brain areas typically associated with deliberation could result from the elicitation of different degrees of cognitive effort involved in decisions regarding sacred versus non-sacred values. As suggested by previous literature, decisions regarding sacred values may rely on deontic rights and wrongs, whereas decisions over non-sacred values may rely on cost–benefit ponderation [31,33]. This would involve different degrees of cognitive effort in both situations: sacred values would work as a heuristic making decisions easy to solve, or cached-offline, whereas decisions regarding non-sacred values would involve some degree of calculation. Such a distinction is consistent with our results.

In addition, we also found that willingness to fight and die ratings were influenced by peers' opinions. In previous studies, sacred value adherence was shown resistant to social influence [8,31], whereas in our study community feedback shifted willingness to fight and die ratings in the direction established by peers. Change in judgement predicted neural activity in the right dlPFC during conflicting feedback for sacred versus non-sacred values, which survived masking with the right dlPFC cluster found in the non-sacred versus sacred value contrast in the first study. The fact that these two right dlPFC clusters overlap provides evidence that individuals who most changed their willingness to fight and die ratings for sacred values after the social manipulation also recruited neural areas associated with non-sacred value processing to a greater extent during the feedback paradigm. Our findings suggest that even when social network interventions are unlikely to reduce commitment to a sacred value [8,12], they could reduce adherence to violent options.

Previous neuroimaging studies have shown vlPFC activation during sacred value processing [31,33]. Yet, we found decreased dlPFC activation. This may owe to the difference in our paradigms. Whereas previous studies examined passive viewing of (non-)sacred values, our paradigm asked participants to evaluate their willingness to fight and die (WFD) for these values. In addition, we found that change in WFD due to false consensus feedback was correlated with activity in the dlPFC. This result is consistent with studies on the role of dlPFC in attitude change. The dlPFC has been shown to play a role in behavioural adjustment after the experience of conflict [43–45]. The role of the dlPFC in conflict resolution is evident in research on adjustment of preferences motivated by cognitive dissonance reduction [46]. This research is consistent with our findings, which indicate that the dlPFC could play a role in social conformity of WFD in the face of peers-conflicting feedback.

We might find somewhat similar results in less radical individuals making extreme statements (which would be an interesting control in a subsequent study); however, the focus of this study was not on identifying distinctive thinking styles that may distinguish radicalized from non-radicalized individuals (which would require studying dimensions of radicalization, such as nationality, radicalization topic, strength of commitment, age grades, gender). Rather, our aim was to uncover the neural mechanisms underlying the distinction between sacred and non-sacred values in a population of interest that expresses support for extreme action in defence of radical ideas. Accordingly, as intended, our results only bear on gaining insight into this sort of radical population. Nonetheless, the issue of whether or not populations in different cultural contexts, and with different degrees of radicalization, display similar neurocognitive processing of sacred values remains an important problem for further research.

In addition, note that we assessed expressed willingness to fight and die for sacred values but not actual behaviour. Yet, we were able to show that willingness to fight and die for sacred values is distinctly articulated by neural mechanisms common to subjective value computation, deontic-rule processing and action control. This appears to rule out posturing in measured expressions of willingness to fight and die, although future studies should address actual causal linkages between sacred values and costly sacrifices both under the fMRI scanner and in association with verifiable

engagement in violence. Finally, although conformity manipulations in laboratory experiments have a rich tradition [25], they may be somewhat artificial when compared with how social feedback influences behaviour in real-world interactions. Future research is thus necessary to determine whether social feedback in more natural settings regarding the use of violence to defend sacred values has similar effects to those we found here.

In line with previous studies [31,33,47], the present findings reveal neural underpinnings of strongly held values in the context of political and religious beliefs. These results may add to understanding of the way commitments to sacred values can be associated with violent activism within the context of intergroup conflict. The question of whether values acquire what we term sacred status (immunity from material trade-offs) because they are important or whether other processes are involved remains a relevant topic for future work.

Ethics. The behavioural studies and neuroimaging sessions were conducted in accordance with IRB Protocol #2014-0926, 'The Neural Basis of Personal Beliefs: A Magnetic Resonance Imaging,' after initial approval by Artis Research IRB00007516 and subsequent review by the U.S. Air Force Surgeon General's Human and Animal Research Panel prior to release of funds in support of this project under AFOSR grant AFOSR FA9550-14-1-0030 DEF to Artis Research. Data protection was a priority in this study and complete anonymity was guaranteed. The researchers were aware of the possible risks related to the self-identification of the participants as supporters of militant jihadi groups. Therefore, we ensured the right of the participants to remain anonymous. The participants were assigned an identity code consisting of five random numbers. All information was stored in a confidential manner and in accordance with the Rules of the UNESCO International Council for Global Health Progress (ICGHP) and followed the Directive 95/46/EC of the European Parliament and of the Council of 24 October 1995 on the protection of individuals with regard to the processing of personal data and on the free movement of such data. All information was kept in password-protected servers and will be destroyed after the publications related to the study. The structure of the database guaranteed the participants' anonymity. We provided the participants with this information before entering the study.

Permission to carry out fieldwork. No licences were required to conduct fieldwork. All fieldwork was approved by the IRB as stated in the Ethics section. All researchers who interacted with participants received certified training in human subjects research protections.

Data accessibility. Our data are deposited in the Dryad Digital Repository: https://doi.org/10.5061/dryad.c0k33vf [48].

Authors' contributions. The project director was S.A. The field studies were conceived by S.A., and they were designed by N.H. and S.A. with the participation of J.G., H.S., A.G. and R.D. Ethnographic and behavioural data were collected by N.H., and analysed by N.H., H.S. and S.A. The fMRI studies were directed by O.V., and were designed by C.P., O.V., N.H. and M.J.C., with the participation of S.A., A.T. and S.C. The fMRI studies were carried out by C.P. and O.V., and the data analysed by C.P., M.J.C., and O.V. The manuscript was written by C.P., O.V., S.A., J.G., M.J.C. and N.H. All authors reviewed the final manuscript. N.H. and C.P. contributed equally to this paper. S.A. and O.V. are the corresponding authors.

Competing interests. The authors declare no competing interests.

Funding. Funding support for this study came from the Minerva Program and the Air Force Office of Scientific Research of the U.S. Department of Defense (http://www.wpafb.af.mil/afrl/afosr/) (AFOSR FA9550-14-1-0030 DEF), and from the BIAL Foundation (no. 163/14). The funders had no role in study design, data collection and analysis, decision to publish, or preparation of the manuscript.

Acknowledgements. We thank everyone who participated in this study.

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
