## [Reviewer comments · Royal Society Open Science]

Review History

RSOS-181585.R0 (Original submission)

Review form: Reviewer 1 (Patricia Churchland)

Is the manuscript scientifically sound in its present form?

Yes

Are the interpretations and conclusions justified by the results?

Yes

Is the language acceptable?

Yes

Is it clear how to access all supporting data?

Yes

Do you have any ethical concerns with this paper?

No

Have you any concerns about statistical analyses in this paper?

No

Recommendation?

Accept as is

Comments to the Author(s)

This is an impressive paper on a difficult but important topic. The experiment was admirably well-thought out and, one must infer, heroically organized.

Of course interpreting the scanning results is filled with uncertainty until we know much more about the network level. The regions of interest (dlPFC, insult, amygdala, anterior consulate etc.) are tagged in many experiments, and we do not really understand exactly what these regions do. Nevertheless the scanning interpretations provided in this paper seem reasonably qualified given the uncertainties. I was puzzled by one thing: the scared values experiment performed by Berns et al 2012 (referred to in the bib) found an increase in dlPFC activity to scared value matters, whereas this paper found a decrease. Perhaps the differences in the experiments (e.g. country of origin of the sample) account for this differences.

The authors are right to mention that it would be very useful to know the differences in scan results that include controls, such as those who are radicalized but do not have strong sacred values, and those who are not radicalized and do not have sacred values. This would be interesting especially because basic personality differences in such matters as being outgoing or being anxious appear to play a significant role in strong political attitudes. But the authors were wise not to try to undertake that experiment this time. So this is not a criticism, just an acknowledgement of their reasonable caution.

Review form: Reviewer 2

Is the manuscript scientifically sound in its present form?

Yes

Are the interpretations and conclusions justified by the results?

Yes

Is the language acceptable?

Yes

Is it clear how to access all supporting data?

Yes

Do you have any ethical concerns with this paper?

No

Have you any concerns about statistical analyses in this paper?

No

Recommendation?

Accept with minor revision (please list in comments)

Comments to the Author(s)

I've read with interest the paper submitted by Hamid and colleagues. This study reports a functional neuroimaging study designed to examine brain processes associated with "costly" decisions involving "sacred values" and whether those sacred values are implemented in similar or distinct circuits.

The very idea that sacred value may be processed differently than non-sacred values is intriguing because a large body of work in cognitive neuroscience and neuroeconomics does not support the idea that social and non-social decision making involve distinct computations, including the vmPFC. I don't think that fMRI has the spatial resolution to tell us this. Even using MVPA doesn't change the fact that one voxel includes hundreds of thousands of neurons. It is possible that Royal Society Open Science journal has a strict word limit (I am not familiar with this outlet). If this isn't the case, I would suggest the authors unpack the paragraph on pg. 5, lines 88-94. The way it is written (too concisely), sounds like a bit neo-phrenology and heavily relying on reverse inference. Similarly, the mention of social influence and brain mechanisms (pg. 6, lines 37-39) is a bit simplistic. Again, that may be due to word-constraint from the journal requirement.

The study design and statistical analysis are sound. The authors took a very conservative approach to try and rule out alternative explanations for the fMRI data (i.e. neurosynth masks, follow-up subjective ratings). I am not a fan of dual process models, but their results are at least consistent with a model where decision-making about non-sacred values (compared to decision-making about sacred values) relies more on cognitive control.

It's a bit surprising that no clusters were more active for sacred value decision-making, and the authors could speculate on why that might be the case. For instance, if they really want to make a dual process argument a la Greene, then the sacred value decisions should elicit response in "affective" regions, e.g. amygdala.

The social conformity effect on willingness to fight and die is interesting, and the authors did a good job highlighting that decisions about sacred values are distinct from the values themselves.

Finally, the authors very briefly mention MVPA (line 292), and then only talk about it again in the results (page 19). It seems like they are only using that to show *some* effect of self-reported moral outrage, but it seems unnecessary.

Review form: Reviewer 3

Is the manuscript scientifically sound in its present form?

Yes

Are the interpretations and conclusions justified by the results?

Yes

Is the language acceptable?

Yes

Is it clear how to access all supporting data?

Yes

Do you have any ethical concerns with this paper?

No

Have you any concerns about statistical analyses in this paper?

No

Recommendation?

Major revision is needed (please make suggestions in comments)

Comments to the Author(s)

This manuscript describes an fMRI study of sacred values in a population of radicalized Islamic adults in Spain. The study aims to investigate the neural mechanisms involved in making decisions about “sacred” versus non-sacred values and also attitude change via peer influence. The involvement of this special population makes this paper’s contribution potentially very important, the experiment is well-conducted, the paper is well-written, and the results are interesting. Nevertheless, the study faces some difficult interpretational issues that could be handled with more care in the manuscript.

The main findings are that making decisions about non-sacred values activated DLPFC, IFC, and parietal areas more than making decisions about sacred values. Interestingly, participants ratings of willingness to fight and die were susceptible to peer influence, for both sacred and non-sacred values. Change in these ratings was associated with activity in DLPFC.

The main difficulty, as I see it, is in interpreting just what exactly these brain activations are reflecting. This is really a general challenge with mapping something a very high-level psychological concept like sacred values onto neural systems. Sacred and non-sacred values differ in several ways. The authors address this by including analyses where several factors such as emotion, familiarity, and salience are regressed out, subtracting away many of the brain regions from the sacred/non-sacred contrast. I think the inclusion of these analyses strengthens the paper and is very important. However, it seems to me that there is a deep issue here as to which of these components is essential to “sacred” cognition and which are ancillary, and it’s not clear to me what the authors are trying to say with these analyses. Surely the emotional component of sacredness is integral, for example. By regressing it out are the authors trying to say that what is left over is the essence of sacredness, or do they believe that all of these components interact together to produce sacred cognition? After all sacredness is correlated with all of these variables so it’s not clear that such a target could be triangulated in this way. It would seem artificial, for example, to mask out regions involved in feeling and reward anticipation if those are integral to the process of cost-benefit analysis.

The one trivial explanation of these results that would probably be the least interesting is that the differences all boil down to cognitive effort. Response times are faster to sacred values compared with non-sacred values, and when we look at these brain maps we may just be seeing maps of decision-making effort that would look similar regardless of the content of the decision. Even though efforts were made to choose non-sacred topics with which the participants were highly familiar, the answers to questions regarding sacred value decisions may be so rote that they do not require any attentional resources to arrive at, they are “cached” off-line so to speak. Is it surprising that there are no brain regions that are more active for sacred decisions than for non-sacred decisions if the difference between the two is a difference in kind of cognitive process instead of just the difference in degree of cognitive processing? I think a deeper discussion of this possibility would strengthen the paper.

There is also not much discussion of how these results relate to other results found in the (admittedly small) neuroimaging of sacred values literature. Discussion of the specifics of this

study's results in light of previous literature seems somewhat cursory and could be expanded to improve the paper. For example, is the activity related to attitude change in the DLPFC consistent with the neuroimaging literature on belief and attitude change?

Etc:

Line 292 refers to MVPA methods in the supporting information, but I do not see any reference to MVPA methods in the supporting materials document.

Decision letter (RSOS-181585.R0)

20-Feb-2019

Dear Professor Atran,

The editors assigned to your paper ("Neuroimaging "will to fight" for sacred values: An empirical case study with supporters of an Al Qaeda affiliate") have now received comments from reviewers, though we offer apologies for the unusual time taken to reach this decision, which was a consequence of difficulty in finding suitable referees. We would like you to revise your paper in accordance with the referee and Associate Editor suggestions which can be found below (not including confidential reports to the Editor). Please note this decision does not guarantee eventual acceptance.

Please submit a copy of your revised paper before 15-Mar-2019. Please note that the revision deadline will expire at 00.00am on this date. If we do not hear from you within this time then it will be assumed that the paper has been withdrawn. In exceptional circumstances, extensions may be possible if agreed with the Editorial Office in advance. We do not allow multiple rounds of revision so we urge you to make every effort to fully address all of the comments at this stage. If deemed necessary by the Editors, your manuscript will be sent back to one or more of the original reviewers for assessment. If the original reviewers are not available, we may invite new reviewers.

If your study uses humans or animals please include details of the ethical approval received, including the name of the committee that granted approval. For human studies please also detail

whether informed consent was obtained. For field studies on animals please include details of all permissions, licences and/or approvals granted to carry out the fieldwork.

- Data accessibility

If you wish to submit your supporting data or code to Dryad (<http://datadryad.org/>), or modify your current submission to dryad, please use the following link:
<http://datadryad.org/submit?journalID=RSOS&manu=RSOS-181585>

- Competing interests

- Authors' contributions

- Acknowledgements

- Funding statement

Kind regards,
Andrew Dunn
Royal Society Open Science Editorial Office

on behalf of Dr Jonathan Roiser (Associate Editor) and Antonia Hamilton (Subject Editor)
openscience@royalsociety.org

Comments to Author:

Reviewers' Comments to Author:

Reviewer: 1

Comments to the Author(s)

This is an impressive paper on a difficult but important topic. The experiment was admirably well-thought out and, one must infer, heroically organized.

Of course interpreting the scanning results is filled with uncertainty until we know much more about the network level. The regions of interest (dlPFC, insult, amygdala, anterior cingulate etc.) are tagged in many experiments, and we do not really understand exactly what these regions do. Nevertheless the scanning interpretations provided in this paper seem reasonably qualified given the uncertainties. I was puzzled by one thing: the scared values experiment performed by Berns et al 2012 (referred to in the bib) found an increase in dlPFC activity to scared value matters, whereas this paper found a decrease. Perhaps the differences in the experiments (e.g. country of origin of the sample) account for this differences.

The authors are right to mention that it would be very useful to know the differences in scan results that include controls, such as those who are radicalized but do not have strong sacred values, and those who are not radicalized and do not have sacred values. This would be interesting especially because basic personality differences in such matters as being outgoing or being anxious appear to play a significant role in strong political attitudes. But the authors were wise not to try to undertake that experiment this time. So this is not a criticism, just an acknowledgement of their reasonable caution.

Reviewer: 2

Comments to the Author(s)

I've read with interest the paper submitted by Hamid and colleagues. This study reports a functional neuroimaging study designed to examine brain processes associated with "costly" decisions involving "sacred values" and whether those sacred values are implemented in similar or distinct circuits.

The very idea that sacred value may be processed differently than non-sacred values is intriguing because a large body of work in cognitive neuroscience and neuroeconomics does not support the idea that social and non-social decision making involve distinct computations, including the vmPFC. I don't think that fMRI has the spatial resolution to tell us this. Even using MVPA doesn't change the fact that one voxel includes hundreds of thousands of neurons. It is possible that Royal Society Open Science journal has a strict word limit (I am not familiar with this outlet). If this isn't the case, I would suggest the authors unpack the paragraph on pg. 5, lines 88-94. The way it is written (too concisely), sounds like a bit neo-phrenology and heavily relying on reverse inference. Similarly, the mention of social influence and brain mechanisms (pg. 6, lines 37-39) is a bit simplistic. Again, that may be due to word-constraint from the journal requirement.

The study design and statistical analysis are sound. The authors took a very conservative

approach to try and rule out alternative explanations for the fMRI data (i.e. neurosynth masks, follow-up subjective ratings). I am not a fan of dual process models, but their results are at least consistent with a model where decision-making about non-sacred values (compared to decision-making about sacred values) relies more on cognitive control.

It's a bit surprising that no clusters were more active for sacred value decision-making, and the authors could speculate on why that might be the case. For instance, if they really want to make a dual process argument a la Greene, then the sacred value decisions should elicit response in "affective" regions, e.g. amygdala.

The social conformity effect on willingness to fight and die is interesting, and the authors did a good job highlighting that decisions about sacred values are distinct from the values themselves.

Finally, the authors very briefly mention MVPA (line 292), and then only talk about it again in the results (page 19). It seems like they are only using that to show *some* effect of self-reported moral outrage, but it seems unnecessary.

Reviewer: 3

Comments to the Author(s)

This manuscript describes an fMRI study of sacred values in a population of radicalized Islamic adults in Spain. The study aims to investigate the neural mechanisms involved in making decisions about "sacred" versus non-sacred values and also attitude change via peer influence. The involvement of this special population makes this paper's contribution potentially very important, the experiment is well-conducted, the paper is well-written, and the results are interesting. Nevertheless, the study faces some difficult interpretational issues that could be handled with more care in the manuscript.

The main findings are that making decisions about non-sacred values activated DLPFC, IFC, and parietal areas more than making decisions about sacred values. Interestingly, participants ratings of willingness to fight and die were susceptible to peer influence, for both sacred and non-sacred values. Change in these ratings was associated with activity in DLPFC.

The main difficulty, as I see it, is in interpreting just what exactly these brain activations are reflecting. This is really a general challenge with mapping something a very high-level psychological concept like sacred values onto neural systems. Sacred and non-sacred values differ in several ways. The authors address this by including analyses where several factors such as emotion, familiarity, and salience are regressed out, subtracting away many of the brain regions from the sacred/non-sacred contrast. I think the inclusion of these analyses strengthens the paper and is very important. However, it seems to me that there is a deep issue here as to which of these components is essential to "sacred" cognition and which are ancillary, and it's not clear to me what the authors are trying to say with these analyses. Surely the emotional component of sacredness is integral, for example. By regressing it out are the authors trying to say that what is left over is the essence of sacredness, or do they believe that all of these components interact together to produce sacred cognition? After all sacredness is correlated with all of these variables so it's not clear that such a target could be triangulated in this way. It would seem artificial, for example, to mask out regions involved in feeling and reward anticipation if those are integral to the process of cost-benefit analysis.

The one trivial explanation of these results that would probably be the least interesting is that the differences all boil down to cognitive effort. Response times are faster to sacred values compared with non-sacred values, and when we look at these brain maps we may just be seeing maps of

decision-making effort that would look similar regardless of the content of the decision. Even though efforts were made to choose non-sacred topics with which the participants were highly familiar, the answers to questions regarding sacred value decisions may be so rote that they do not require any attentional resources to arrive at, they are “cached” off-line so to speak. Is it surprising that there are no brain regions that are more active for sacred decisions than for non-sacred decisions if the difference between the two is a difference in kind of cognitive process instead of just the difference in degree of cognitive processing? I think a deeper discussion of this possibility would strengthen the paper.

There is also not much discussion of how these results relate to other results found in the (admittedly small) neuroimaging of sacred values literature. Discussion of the specifics of this study’s results in light of previous literature seems somewhat cursory and could be expanded to improve the paper. For example, is the activity related to attitude change in the DLPFC consistent with the neuroimaging literature on belief and attitude change?

Etc:

Line 292 refers to MVPA methods in the supporting information, but I do not see any reference to MVPA methods in the supporting materials document.

Author's Response to Decision Letter for (RSOS-181585.R0)

See Appendix A.

RSOS-181585.R1 (Revision)

Review form: Reviewer 2

Is the manuscript scientifically sound in its present form?

Yes

Are the interpretations and conclusions justified by the results?

Yes

Is the language acceptable?

Yes

Is it clear how to access all supporting data?

Yes

Do you have any ethical concerns with this paper?

No

Have you any concerns about statistical analyses in this paper?

No

Recommendation?

Accept as is

Comments to the Author(s)

The authors did a great job with their revision

Review form: Reviewer 3

Is the manuscript scientifically sound in its present form?

Yes

Are the interpretations and conclusions justified by the results?

Yes

Is the language acceptable?

Yes

Is it clear how to access all supporting data?

Yes

Do you have any ethical concerns with this paper?

No

Have you any concerns about statistical analyses in this paper?

No

Recommendation?

Accept as is

Comments to the Author(s)

I believe the authors have addressed my concerns adequately. The issues I raised were largely interpretational, and the authors have expanded the interpretation of their results and the discussion of how they relate to previous literature. I agree with their decision to eliminate the MVPA results from the analysis.

Decision letter (RSOS-181585.R1)

03-May-2019

Dear Professor atran,

I am pleased to inform you that your manuscript entitled "Neuroimaging "will to fight" for sacred values: An empirical case study with supporters of an Al Qaeda affiliate" is now accepted for publication in Royal Society Open Science.

on behalf of Dr Jonathan Roiser (Associate Editor) and Antonia Hamilton (Subject Editor)
openscience@royalsociety.org

Reviewer comments to Author:

Reviewer: 2

Comments to the Author(s)

The authors did a great job with their revision

Reviewer: 3

Comments to the Author(s)

I believe the authors have addressed my concerns adequately. The issues I raised were largely interpretational, and the authors have expanded the interpretation of their results and the discussion of how they relate to previous literature. I agree with their decision to eliminate the MVPA results from the analysis.

Follow Royal Society Publishing on Twitter: [@RSocPublishing](https://twitter.com/RSocPublishing)

Appendix A

We would like to thank all of the reviewers for their thoughtful comments. And we would like to extend a further appreciation to the editors for affording us this opportunity to improve our manuscript with the help of reviewer insights.

Reviewers' Comments to Author:

Reviewer: 1

Comments to the Author(s)

This is an impressive paper on a difficult but important topic. The experiment was admirably well-thought out and, one must infer, heroically organized.

Of course interpreting the scanning results is filled with uncertainty until we know much more about the network level. The regions of interest (dlPFC, insula, amygdala, anterior cingulate etc.) are tagged in many experiments, and we do not really understand exactly what these regions do. Nevertheless the scanning interpretations provided in this paper seem reasonably qualified given the uncertainties.

1. I was puzzled by one thing: the scared values experiment performed by Berns et al 2012 (referred to in the bib) found an increase in dlPFC activity to scared value matters, whereas this paper found a decrease. Perhaps the differences in the experiments (e.g. country of origin of the sample) account for this differences.

We thank the reviewer for his/her comments. To clarify: Berns et al 2012 identified the ventrolateral prefrontal cortex (vlPFC) in response to sacred values, and we found decreased dorsolateral prefrontal cortex (dlPFC) activity. These two regions are associated with different functions. On one hand, as mentioned in Berns et al 2012, the vlPFC has been typically associated with rule retrieval. On the other hand, the dlPFC is associated with cognitive control and deliberation. The decrease in dlPFC that we identified does not overlap with Bern's vlPFC, it's a functionally distinct area.

Perhaps another question is why didn't we find vlPFC as did Berns et al 2012? A reasonable explanation is that we employed a different experimental paradigm. That is, Berns et al 2012 found vlPFC in response to **passive viewing of sacred values**, while in our paradigm participants had to convey their **willingness to fight and die for sacred values**, which activates decision-making mechanisms in the brain. If we had used a passive viewing paradigm comparing sacred and non-sacred value processing, we may have been able to detect vlPFC activity associated with sacred values.

We have added this interpretation to the discussion section with the following sentences (pp. 23, lines 559-563):

“Previous neuroimaging studies have shown vlPFC activation during sacred value processing (31,33). Yet, we found decreased DLPFC activation. This may owe to the

difference in our paradigms. Whereas previous studies examined passive viewing of (non-)sacred values, our paradigm asked participants to evaluate their willingness to fight and die for these values.”

2. The authors are right to mention that it would be very useful to know the differences in scan results that include controls, such as those who are radicalized but do not have strong sacred values, and those who are not radicalized and do not have sacred values. This would be interesting especially because basic personality differences in such matters as being outgoing or being anxious appear to play a significant role in strong political attitudes. But the authors were wise not to try to undertake that experiment this time. So this is not a criticism, just an acknowledgement of their reasonable caution.

We agree with the reviewer about including a control group in future studies, although the issue of what could count as selection criteria for controls merits its own theoretical discussion.

Reviewer: 2

Comments to the Author(s)

I’ve read with interest the paper submitted by Hamid and colleagues. This study reports a functional neuroimaging study designed to examine brain processes associated with “costly” decisions involving “sacred values” and whether those sacred values are implemented in similar or distinct circuits.

1. The very idea that sacred value may be processed differently than non-sacred values is intriguing because a large body of work in cognitive neuroscience and neuroeconomics does not support the idea that social and non-social decision making involve distinct computations, including the vmPFC. I don’t think that fMRI has the spatial resolution to tell us this. Even using MVPA doesn’t change the fact that one voxel includes hundreds of thousands of neurons.

We thank the reviewer for his/her comment. We agree with the reviewer that fMRI alone cannot prove that SVs are processed differently than nSVs. Nevertheless, fMRI has been useful in many domains, and with several different experimental paradigms, to support theoretical hypotheses about how humans process, for example, beliefs. Even if the spatial resolution of fMRI is, in fact, at multivoxel level, it is also true that different regions of the brain, such as the vmPFC or the DLPFC, have been systematically associated with different functions.

2. It is possible that Royal Society Open Science journal has a strict word limit (I am not familiar with this outlet). If this isn't the case, I would suggest the authors unpack the paragraph on pg. 5, lines 88-94. The way it is written (too concisely), sounds like a bit neo-phrenology and heavily relying on reverse inference.

Following the reviewer's advice, we have edited this paragraph as follows (pg. 5, lines 89-100):

“Previous work has identified brain regions implicated in utilitarian cost-benefit calculations during value-based decision-making. Specifically, trading off costs and benefits -- both in simple decisions involving outcomes for oneself as well as more complex moral decisions -- has been associated with neural activity in subcortical regions, such as striatum and amygdala, as well as cortical regions including lateral prefrontal cortex, ventromedial prefrontal cortex and posterior parietal cortex (20,21). The vmPFC, in particular, has been hypothesized to function as a global value comparator during value-based choices, and its activity has been found to correlate with value differences between chosen and unchosen items in a variety of contexts, from monetary settings up to food preferences (20, 22). In addition, prior studies suggest that the dlPFC modulates vmPFC activity during decisions that require cognitive control, such as sticking to healthy food choices (23), or reweighting of stimulus in front of new contextual information in monetary settings (24).”

3. Similarly, the mention of social influence and brain mechanisms (pg. 6, lines 37-39) is a bit simplistic. Again, that may be due to word-constraint from the journal requirement.

We agree that our mention of social influence and brain mechanisms was cursory. Accordingly, we've extended our section on this topic as follows (pp. 6-7, lines 136-156):

“In our second fMRI experiment we investigated one such strategy: influencing via peer feedback. Ever since Solomon Asch's (25) pioneering work on social conformity, extensive research in social cognition has shown that human behaviour is not only guided by subjective attitudes and values but also by perceptions of others' attitudes and values (26). Some studies have shown that inducing false perceptions of others' moral judgments can influence people to change their own moral judgments (27,28). Research on sacred values, however, has shown that such values are resistant to social influence (8). Neuroimaging research suggests that when social consensus conflicts with subjects' judgments, the degree of activation in prediction error networks and orbitofrontal cortex predicts the degree of change in judgments at a later time (29,30). The only neuroimaging study published on social conformity and sacred values to-date revealed that increased VLPFC activity for sacred values predicted weaker

social conformity, whereas decreased VLPFC activity predicted greater conformist behavior (31).

Research on radicalization distinguishes between deradicalization and disengagement, suggesting that former violent extremists rarely change their beliefs (deradicalize) but more often lose their motivation to defend them (disengage) (32). Accordingly, we conjectured that it might be possible to induce flexibility in the way people defend their sacred values. Thus, we tested whether willingness to fight and die for sacred values is amenable to social influence and, if so, whether regions previously associated with attitude change (29,30) correlate with change in judgment for sacred versus non-sacred values.”

4. The study design and statistical analysis are sound. The authors took a very conservative approach to try and rule out alternative explanations for the fMRI data (i.e. neurosynth masks, follow-up subjective ratings). I am not a fan of dual process models, but their results are at least consistent with a model where decision-making about non-sacred values (compared to decision-making about sacred values) relies more on cognitive control.

We appreciate the reviewer’s caution about dual-process models, and this why – as the reviewer indicates – we took a conservative approach to add rigor to our design and ensure the plausibility of our model if, as indeed it turned out, the model was supported by robust statistical results.

5. It’s a bit surprising that no clusters were more active for sacred value decision-making, and the authors could speculate on why that might be the case. For instance, if they really want to make a dual process argument a la Greene, then the sacred value decisions should elicit response in “affective” regions, e.g. amygdala.

We agree with the reviewer that the lack of areas with increased activation for sacred values is intriguing. In response to the reviewer’s observation we have added the following paragraph to our discussion to explore possible reasons for why our data did not show positive activity for sacred values (pg. 22, lines 533-537):

“The question remains of why there were no brain regions associated with affective processing, such as the amygdala, which activated during the sacred compared to the non-sacred value condition. We believe that the most likely explanation for this owes to our experimental paradigm not being sensitive enough to detect the differential neural activity associated with affective regions.”

6. The social conformity effect on willingness to fight and die is interesting, and the authors did a good job highlighting that decisions about sacred values are distinct from the values themselves.

We thank the reviewer for his/her comment.

7. Finally, the authors very briefly mention MVPA (line 292), and then only talk about it again in the results (page 19). It seems like they are only using that to show *some* effect of self-reported moral outrage, but it seems unnecessary.

We agree that the MVPA analysis could be seen as redundant and therefore have eliminated all mention of it from the manuscript.

Reviewer: 3

Comments to the Author(s)

This manuscript describes an fMRI study of sacred values in a population of radicalized Islamic adults in Spain. The study aims to investigate the neural mechanisms involved in making decisions about “sacred” versus non-sacred values and also attitude change via peer influence. The involvement of this special population makes this paper’s contribution potentially very important, the experiment is well-conducted, the paper is well-written, and the results are interesting. Nevertheless, the study faces some difficult interpretational issues that could be handled with more care in the manuscript.

The main findings are that making decisions about non-sacred values activated DLPFC, IFC, and parietal areas more than making decisions about sacred values. Interestingly, participants ratings of willingness to fight and die were susceptible to peer influence, for both sacred and non-sacred values. Change in these ratings was associated with activity in DLPFC.

1. The main difficulty, as I see it, is in interpreting just what exactly these brain activations are reflecting. This is really a general challenge with mapping something a very high-level psychological concept like sacred values onto neural systems. Sacred and non-sacred values differ in several ways. The authors address this by including analyses where several factors such as emotion, familiarity, and salience are regressed out, subtracting away many of the brain regions from the sacred/non-sacred contrast. I think the inclusion of these analyses strengthens the paper and is very important. However, it seems to me that there is a deep issue here as to which of these components is essential to “sacred” cognition and which are ancillary, and it’s not clear to me what the authors are trying to say with these analyses. Surely the emotional component of sacredness is integral, for example. By regressing it out are the authors trying to say that what is left over is the essence of sacredness, or do they believe that all of these components interact together to produce sacred cognition? After all sacredness is correlated with all of these variables so it’s not clear that such a target could be triangulated in this way. It would seem artificial, for example, to

mask out regions involved in feeling and reward anticipation if those are integral to the process of cost-benefit analysis.

We agree with the reviewer that emotion, as well as other factors correlated with sacred values, may be an integral part of sacred values. However, we sought to ensure that sacredness could not be attributed to any one of these factors alone. Given our paradigm, the best way to verify this would be to regress out these factors. By regressing out emotional intensity, familiarity, and salience, we provide evidence that sacred value processing relies on brain areas that do not overlap with brain regions associated with each one of these factors. This does not imply that the brain areas associated with emotion and the other controlled factors are not typically active during sacred value processing; rather, it suggests that sacred values rely on more than just those areas. Sacred values may well have inherent emotional dimensions; however, our study indicates that sacred values are associated with cognitive factors beyond emotional processes. We hope that this study contributes to further theorizing about the components of sacredness.

In response to the reviewer's concerns, we have added the following sentence to the discussion (pg. 18, lines 436-438):

“This is not to deny that emotions and other factors may be integral to sacred values (5); however, our approach sought to ensure that sacredness could not be attributed exclusively to these other factors.”

2. The one trivial explanation of these results that would probably be the least interesting is that the differences all boil down to cognitive effort. Response times are faster to sacred values compared with non-sacred values, and when we look at these brain maps we may just be seeing maps of decision-making effort that would look similar regardless of the content of the decision. Even though efforts were made to choose non-sacred topics with which the participants were highly familiar, the answers to questions regarding sacred value decisions may be so rote that they do not require any attentional resources to arrive at, they are “cached” off-line so to speak. Is it surprising that there are no brain regions that are more active for sacred decisions than for non-sacred decisions if the difference between the two is a difference in kind of cognitive process instead of just the difference in degree of cognitive processing? I think a deeper discussion of this possibility would strengthen the paper.

The reviewer makes an interesting point about the neural differences between sacred and non-sacred values perhaps resulting from differences in cognitive effort. We believe what's interesting about our results is that we found differences in cognitive effort in areas associated with deliberation. This indicates that these differences may be differences in effort of deliberation, not in cognitive effort generally. Accordingly, we have edited the discussion in the following way (pg. 22-23, lines 538-546):

“The finding that non-sacred compared to sacred value processing relied on brain areas typically associated with deliberation could result from the elicitation of different degrees of cognitive effort involved in decisions regarding sacred versus non-sacred values. As suggested by previous literature, decisions regarding sacred values may rely on deontic rights and wrongs, whereas decisions over non-sacred values may rely on cost-benefit ponderation (31,33). This would involve different degrees of cognitive effort in both situations: sacred values would work as a heuristic making decisions easy to solve, or cached-offline, whereas decisions regarding non-sacred values would involve some degree of calculation, such a distinction is consistent with our results.”

3. There is also not much discussion of how these results relate to other results found in the (admittedly small) neuroimaging of sacred values literature. Discussion of the specifics of this study’s results in light of previous literature seems somewhat cursory and could be expanded to improve the paper. For example, is the activity related to attitude change in the DLPFC consistent with the neuroimaging literature on belief and attitude change?

We agree that our discussion was too limited on this topic and have added the following paragraph to our discussion section (pp. 23-24, lines 559-569):

“Previous neuroimaging studies have shown vIPFC activation during sacred value processing (31,33). Yet, we found decreased DLPFC activation. This may owe to the difference in our paradigms. Whereas previous studies examined passive viewing of (non-)sacred values, our paradigm asked participants to evaluate their willingness to fight and die for these values. In addition, we found that change in WFD due to false consensus feedback was correlated with activity in DLPFC. This result is consistent with studies on the role of DLPFC in attitude change. DLPFC has been shown to play a role in behavioral adjustment after the experience of conflict (42-44). The role of DLPFC in conflict resolution is evident in research on adjustment of preferences motivated by cognitive dissonance reduction (45). This research is consistent with our findings, which indicate that DLPFC could play a role in social conformity of WFD in the face of peers-conflicting feedback.”

Etc:

4. Line 292 refers to MVPA methods in the supporting information, but I do not see any reference to MVPA methods in the supporting materials document.

Owing to redundancy with the univariate analysis results, we have eliminated the MVPA from the revised manuscript.